# Modeling the Theory of Planned Behaviour to predict adherence to preventive dental visits in preschool children

**Maryam Elyasi**[1], **Hollis Lai**[2], **Paul W. Major**[1], **Sarah R. Baker**[3], **Maryam Amin**[4]*

**1** Orthodontic Graduate Program, School of Dentistry, University of Alberta, Edmonton, Alberta, Canada, **2** School of Dentistry, University of Alberta, Edmonton, Alberta, Canada, **3** Academic Unit of Oral Health, Dentistry and Society, University of Sheffield, Sheffield, United Kingdom, **4** Division of Pediatric Dentistry, University of Alberta, Edmonton, Alberta, Canada

* maryam.amin@ualberta.ca

**Data Availability Statement:** The data has been deposited to "CSIRO Data Access Portal" and can be accessed via: https://www.openicpsr.org/openicpsr/project/116861/version/V1/view/.

## Abstract

### Objectives

Dental caries is the most common chronic childhood disease that occurs in a continuum and can be prevented by children and their parents' adherence to recommended oral health behaviors. Theory-driven tools help practitioners to identify the causes for poor adherence and develop effective interventions. This study examined the Expanded Theory of Planned Behaviour (TPB) Model by adding the concept of Sense of Coherence (SOC) to predict parental adherence to preschooler's preventive dental visits.

### Methods

Data regarding socio-economic demographics were collected from parents of children aged 2–6 years. Constructs of TPB including parental attitudes, subjective norms (SN), Perceived Behavioural Control (PBC), and intention to attend preventive dental visits for their pre-schoolers were collected by questionnaire, alongside parents' sense of coherence (SOC). Dental attendance was measured by asking if the child had a regular dental visit during the last year. Structural Equation Modeling Analysis (SEMA) was carried out to identify significant direct and indirect (mediated) pathways in the extended TPB model.

### Results

Three hundred and seventy-eight mothers (mean age = 34.41 years, range 22–48) participated in the study. The mean age of children was 3.92 years, range: 2–6), and 75.9% had dental insurance. Results of the final model showed that predisposing factors (child's birth-place and mother's birthplace) significantly predicted enabling resources (family monthly income and child's dental insurance status); both predicted the TPB components (PBC, SN, and attitude). TPB components, in turn, predicted behavioural intention. However, contrary to expectation, intention did not significantly predict dental attendance in the past 12 months. Parent's SOC significantly predicted TPB components and dental attendance.

**Funding:** ME & MA received the following sources of funding for this study: 1. Canadian Institutes of Health Research (CIHR) grant (FRN: 126751) 2. Fund for Dentistry, University of Alberta (Project #: 2015-03) 3. Alpha Omega Foundation of Canada, Postdoctoral Education Research Grant 2015-16. 4. American Association of Orthodontics Foundation (AAOF), Resident Research Award 2017.

**Competing interests:** The authors have declared that no competing interests exist.

Overall, 56% of the variance in dental attendance was explained by the expanded TPB model.

## Conclusions

The expanded TPB model explained a great deal of variance in preschooler's dental attendance. These findings suggest that the expanded model could be used as the framework for designing interventions or strategies to enhance dental attendance among preschoolers; in particular, such strategies should focus specifically on enhancing parental SOC including empowerment.

## Introduction

The most common chronic disease in children, dental caries, is almost entirely preventable with adequate adherence to recommended oral health behaviours including good oral hygiene, dietary habits, and regular dental visits [1, 2]. However, more than 40% of children have tooth decay by the time they reach preschool [3]. Canadian Dental Association reported an estimated 2.26 million school-days missed annually in Canada due to dental diseases that account for about one-third of day surgeries for preschoolers aged 1–5 nationwide [4]. Therefore, the prevention of dental caries at younger ages, similar to any other chronic health conditions, could reduce many serious dental problems that would compromise children's general health and well-being and their quality of life over the lifespan [1].

Adherence to a healthy diet (consuming unsweetened foods and beverages) and good oral hygiene practices (tooth brushing twice a day with fluoride) are examples of professional recommendations for preventing dental caries in children [2, 5, 6]. These daily home preventive measures are complemented by attending regular dental visits, which not only allow for early detection and management of oral diseases but also enhance parental awareness of the cause and prevention of the disease [2, 7]. The American Academy of Pediatric Dentistry (AAPD) recommends that children have dental examinations every six months, starting six months after the eruption of the first tooth but no later than their first birthday [5].

Although most studies seldom differentiate children's dental attendance between preventive and restorative visits, adherence to either type of visits have been found to be unsatisfactory [8]. Nearly half of US children do not receive preventive dental visits (as recommended by the AAPD/Bright Futures report), and those younger than six years are the least likely to receive it [8]. With the importance of preventive dental visits for children established, more attention has been paid to adherence to their preventive measures concerning oral hygiene and dietary habits than dental attendance [7]. Few studies have examined parental adherence to these recommendations, and when they did they show inconsistent and conflicting results [2, 7].

Adherence to professional recommendations in chronic health conditions has been recognized as a challenge among health care providers [9]. Parents, especially mothers, have a prominent influence on children's oral health behaviours as professional recommendations include regular dental visits; actions that children cannot independently adhere [10, 11]. Health behaviour theories, therefore, have been used to better understand the determinants of adherence behaviours [11]. Specifically, the Theory of Planned Behaviour (TPB) is a popular explanatory model for preventive health behaviours [12]. According to this theory, behaviour is a function of their intention moderated by Perceived Behavioural Control (PBC). Their intention, in

turn, is influenced by their attitudes toward their behaviour, subjective norms, and PBC [12]. Like attitudes and subjective norms, PBC has an impact on intention. In addition, PBC can also affect their behaviour directly, to the extent that the perception of control accurately reflects actual control [12]. In sum, PBC and intention can be used together to predict behavior.

TPB has been applied in many oral health studies [13–17] and reported as the most frequently used theoretical framework to design theory-based studies in oral health domain [13]. However, its application in children's oral health research is relatively new [11]. In the study by Van den Branden et al. in 2013, the predictive validity of the TPB was examined in relation to oral health behaviours of parents regarding their preschooler's; it was found that the TPB components accounted for 41% to 46% of the variance in predicting annual dental visits and tooth brushing twice a day among 5-year-old children in Belgium [11]. One advantage of the TPB is that it can accommodate the inclusion of additional constructs contributing to the elicitation of a particular behaviour and its predictive properties could, therefore, be enhanced by other variables known to be important in adherence [12], [18].

It is well-known that parental adherence to preventive measures for their children is determined by their ability to cope with daily stressors and to identifying and mobilizing resources to adhere to healthy practices [19]. Although the TPB has elucidated how patients conceptualize health-threatening conditions and evaluate possible facilitators and barriers towards adherence, it does not address behavioural coping skills very well [9]. The ability to deal with life stressors has been examined previously in relation to health through the concept of Sense of Coherence (SOC) [20]. Studies have shown the influence of parents' SOC on children's oral health behaviours [19, 21–24] and oral-health-related quality of life (OHRQoL) [24, 25]. Mothers with higher SOC were more likely to have positive attitudes and behaviours towards their children's oral health than those with lower SOC [21]. Mothers' SOC has also been found to be significantly associated with their children's dental attendance pattern even after adjusting for socioeconomic variables [19, 23]. Although studies showed that the TPB model accounts for predicting parental intention well, the effects of daily stressors on their intention and its transition to behavior are not clear; therefore, SOC is used as a proxy for life stressors in this study to see how this construct can contribute to our TPB model.

Following this path of research, this study aimed to investigate the inclusion of SOC as an expanded TPB model to predict parental adherence to preventive dental visits for their children. We hypothesized that the development of an expanded TPB model would enhance the predictive power of the TPB model in predicting dental attendance behaviour in preschoolers.

## Materials and methods

### Study setting and participants

This multi-center cross-sectional study was granted ethics approval from the University of Alberta Research Ethics Board (Protocol No. 00047287) and Alberta Health Services. A representative sample of English-speaking mothers of children aged 2–6 years living in Edmonton, Alberta, was recruited through vaccination programs in randomly selected community health centers located in four geographical areas in Edmonton.

### Sample size calculation

According to the 2013–14 Alberta Health Services Report, the overall vaccination rate for preschoolers in Edmonton was 91.7%. A representative sample of this population was estimated at 370 participants given the prevalence of adherence to oral health-related behaviours among

Canadian-born children is 72% [26], a marginal error of 5%, 95% Confidence Interval (CI), and 20% possible participant losses.

## Data collection/procedure

A trained research assistant (RA) collected data from four randomly selected community health centers in Edmonton during immunization events for preschoolers. The RA explained the study to mothers in the waiting room and gave them an information letter and consent form. Once a signed consent form was obtained, mothers were asked to complete a questionnaire that included four sections as following and took about 20 minutes to complete.

**a. Socio-demographic characteristics.** Predisposing characteristics including child's gender, age, birthplace (whether in Canada or no), mothers' age and birthplace (whether in Canada or not) as well as enabling resources including child's dental insurance status (yes or no) and type of insurance (public or private), mother's level of education (high school or under, college or university degree), and monthly household income (less than $3,000, $3000 to $5,000, and more than $5,000 CAD) were collected in section one.

**b. Theory of Planned Behaviour questionnaire.** The second section was a 24-item validated questionnaire based on Azjen's Theory of Planned Behaviour (TPB) constructs adopted to examine parental attitudes (8 items), subjective norms (10 items), PBC (5 items), and intention (1 item) towards their preschoolers' dental attendance [11, 27]. Participants rated each item on a 7-point Likert scale ranged from strongly agree to strongly disagree. The responses to the items measuring the TPB constructs were summed to indicate their final scores; therefore, the higher total score for items measuring participants' attitude denoted a more positive attitude.

**c. Sense of Coherence questionnaire.** The third section was a 13-item validated questionnaire for measuring mothers' SOC (SOC-13) based on three concepts including comprehensibility (five items), manageability (four items), and meaningfulness (four items). The response options for each item followed a Likert scale from one to seven and the scores of the negatively worded items were reversed for the analysis so that a higher score for each concept denoted a stronger SOC [20].

**d. Oral health behaviours.** In the last section, mothers' self-reported oral health behaviours of their children were collected; the frequency of sugary food or sugary drink intake was measured in response to the question *'How often does your child consume foods, drinks or snacks high in sugar?*(with binary answers *'never or less than once a day, equal to or greater than once a day')*; oral hygiene was assessed in response to the *question 'How many times a day are your child's teeth cleaned?'* (with binary answers *'less than twice a day, equal to or greater than twice a day')*; the frequency and pattern of dental attendance were evaluated by two questions *'when the child had his/her last dental visit* (with binary answers*'within the last 12 months, over one year or never had one')*; and *'what was (were) the reason(s)?'* (with binary answers *'regular check-up, non-urgent or urgent dental problems')*.

## Statistical analyses

The descriptive characteristics of the participants were explored using SPSS 24.0 software (IBM Corp., Armonk, N.Y., USA). Structural Equation Modelling (SEM) was used to analyze the data. Two-stage SEM is currently the best method for testing prior theoretical models [28]. In the first stage, the measurement model, confirmatory factor analysis (CFA) was used to examine whether the indicators (items) chosen to measure the four latent (underlying) constructs were acceptable. The indicators that have been used are the ones that allow for a best-fitting model. The four latent factors were; predisposing factors (indicators: child's birthplace,

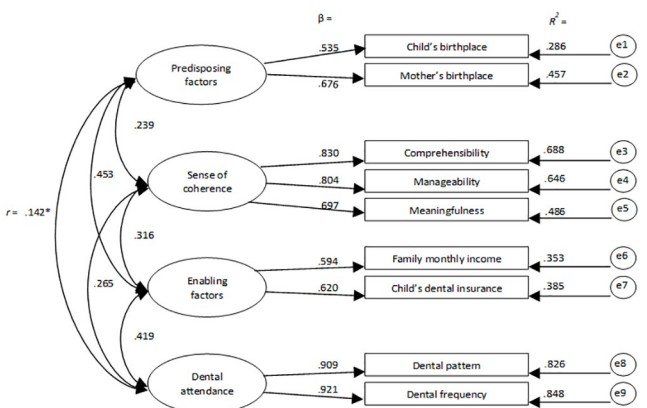

**Fig 1. Bootstrapped ML standardised estimates for the confirmatory factor analysis.** For all pathways $p < 0.01$ except *.

mother's birthplace), enabling factors (indicators: family monthly income, child's dental insurance status), SOC (indicators: comprehensibility, manageability, meaningfulness domains), and dental attendance (indicators: attendance frequency, attendance pattern).

CFA provides information on how indicator items (e.g. child's birthplace) measure underlying (latent) constructs (e.g. predisposing factors). The initial step of the analysis was to test a first-order CFA with predisposing factors, enabling factors, SOC, and dental attendance as the four latent constructs. Scale items (indicators) representing each of the four latent constructs are detailed in Fig 1. Items were not allowed to load on more than one construct nor were error terms allowed to correlate.

Following the specification of the measurement model, the second stage of the analysis was to test a structural model, which examined the direct and indirect relationships between the constructs as hypothesized within the amended TPB model. In accordance with the TPB and with SOC as an additional factor, 27 direct pathways were hypothesized; predisposing factors would predict enabling factors, and both of these would predict the three TPB components (perceived attitude, behavioural control, and subjective norms). The three TPB components would predict perceived behavioural intention, and all would, in turn, predict dental attendance. Predisposing and enabling factors would also predict dental attendance, toothbrushing and sugar intake frequency. Concerning SOC, we hypothesized that it would predict the TPB components, behavioural intention, dental attendance, toothbrushing and sugar intake frequency.

AMOS estimates the total effects, which are made up of both direct effects (a path directly from one variable to another, e.g. predisposing to enabling factors) and indirect effects (a path mediated through other variables, e.g. predisposing → dental attendance via enabling resources). The model was estimated using bootstrapping wherein multiple samples (n = 900+) are randomly drawn from the original sample. The CFA model is then estimated in each data-set, and the results averaged. The ML bootstrap estimates and standard errors [together with bias-corrected 95% confidence intervals (CIs)] are then compared with the results from the original sample to examine the stability of parameters and test statistics [29]. The full model illustrating direct & indirect effects can be seen in Figs 2 and 3.

As recommended, the model fit was evaluated using a range of indices [29, 30]. A χ2/df ratio of <3.0, RMSEA values <0.06, CFI and TLI ≥.9 and a SRMR <0.08 were taken to indicate an acceptable model fit (30).

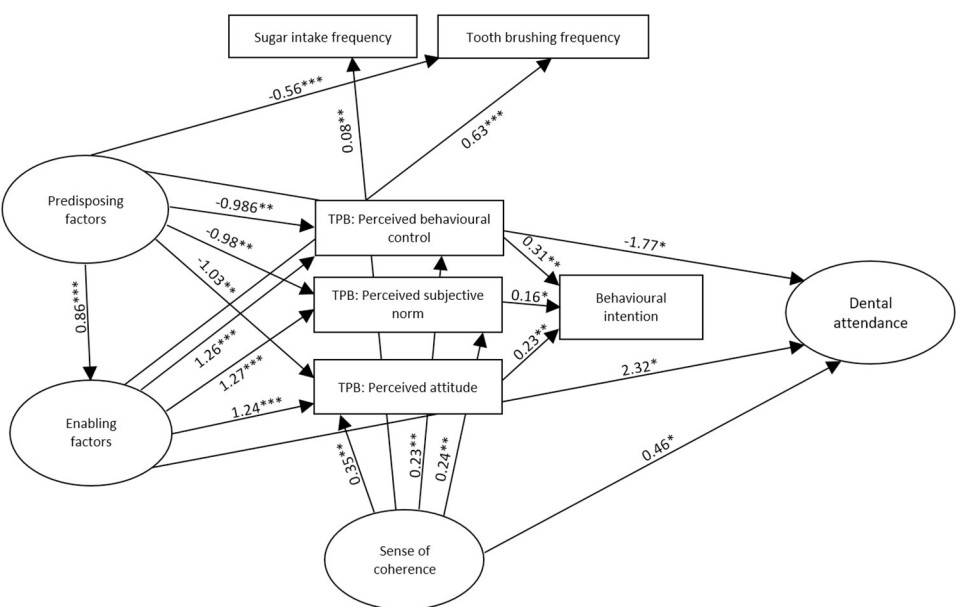

**Fig 2. Bootstrapped standardized direct effect estimates for the amended TPB for dental attendance in preschool children illustrated with solid arrows.** For ease of interpretation, only significant paths shown, and error and indicator variables omitted. * p < .05, ** p < .01, *** p ≤ .001.

## Results

The response rate was 95%. The mean age of 378 mothers who participated in this research was 34.4±4.9 years. All collected data were used in the analysis as there were no outlying results. Among the preschoolers with the mean age of 3.92±1.33 years, 191 (50.6 percent) were

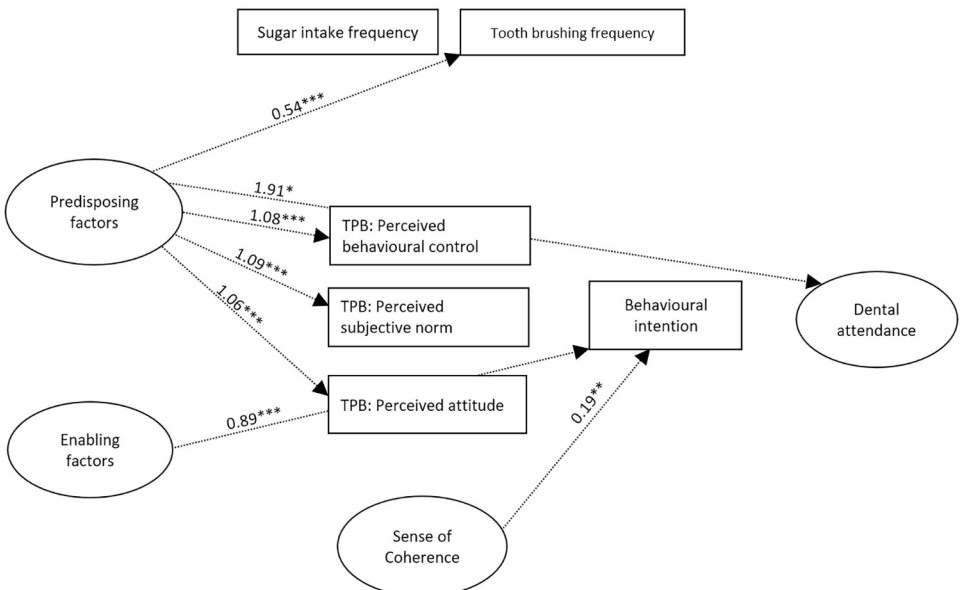

**Fig 3. Bootstrapped standardized indirect effect estimates for the amended TPB for dental attendance in preschool children illustrated with dotted arrows.** For ease of interpretation, only significant paths shown, and error and indicator variables omitted. * p < .05, ** p < .01, *** p ≤ .001.

girls. Participants' characteristics are presented in Table 1. The Cronbach's alpha for the subset of items applied to measure attitude, subjective norm, and PBC were 0.74, 0.83, and 0.76 respectively. The SOC-13 scale showed acceptable internal consistency (Cronbach alpha: 0.91) in this study.

## Confirmatory factor analysis

For the CFA, test of basic assumptions including univariate and multivariate normality, linearity and multi-collinearity were conducted. Logarithmic transformation of data was applied for non-normal data. Testing the specification, identification, and estimation of the model showed an acceptable fit on all a priori indices ($X^2$ = 2.563, SRMR = 0.039, CFI = 0.970, TLI = 0.907, RMSEA = 0.064, Cis = 0.043/0.086). The bootstrapped standardized estimates for this four-factor measurement model is presented in Fig 1. Factors (latent variables) are in ellipses, items (indicator variables) are in rectangles and residual error terms in circles. As seen in Fig 1, all factor loadings were significant and in the expected direction. Both the child and mother being born in Canada were associated with more of the 'predisposing' factor (with factor loadings of 0.54 and 0.68 respectively). Having a higher family income and dental insurance for the child were associated with more of the 'enabling resources' factor. A preventive-orientated attendance and visiting the dentist regularly were associated with more of the 'dental attendance' factor (with factor loadings of 0.91, 0.92 respectively). Greater manageability, comprehensibility and meaningfulness were associated with more sense of coherence. The correlations among the four latent factors ranged between 0.14 and 0.45, indicating that they had acceptable discriminant validity (i.e. <0.85) (Fig 1).

## The extended TPB model

The model was an acceptable fit to the data meeting all apriori indices ($x^2$ = 2.432, SRMR = 0.066, CFI = 0.941, TLI = 0.911, RMSEA = 0.062, CIs = 0.050/0.074). Within this model, eight of the hypothesized bootstrapped paths were non-significant; predisposing and enabling factors to sugar intake frequency, SOC to behavioural intention, SOC to tooth brushing frequency, each of the three TPB components to dental attendance, and behavioural intention to dental attendance. All hypothesized paths within the model are presented in Table 2. The remaining paths were significant and can be seen in Fig 2. The bootstrapped percent of variance accounted for were: enabling factors(73%), attitude(54%), subjective norm(50%), perceived behavioural control (49%), intention (34%) and dental attendance (56%).

**Direct effects.** All of the significant direct paths were in the expected direction (Table 2); more of the predisposing factor was linked to more enabling resources; greater predisposing and enabling resources were linked to higher perceived attitude, subjective norms and PBC scores, to greater dental attendance, and a higher frequency of tooth brushing; A greater SOC was linked to higher perceived attitude, subjective norms and PBC scores, greater dental attendance and less frequent sugar intake (Fig 2). The three TPB components were all linked to a greater behavioural intention but were not, as hypothesized, linked to dental attendance. In addition, surprisingly, the behavioural intention was not associated with greater dental attendance.

**Indirect effects.** There were a number of significant indirect effects between latent and observed variables within the model (Table 2). Predisposing factors were linked indirectly to the TPB components, dental attendance and toothbrushing via enabling factors (Fig 3). It seems that the relationship between the predisposing factor (i.e. child and mother born in Canada) and higher scores on perceived attitude, subjective norms, and perceived control, more frequent toothbrushing as well as a greater dental attendance, may be mediated by a higher

**Table 1. Participants' characteristics.**

| Characteristics | N (%) |
|---|---|
| *Mother's level of education* | |
| High school or under | 83 (21.9%) |
| College or Trade | 149 (39.4%) |
| University degree | 146 (38.6%) |
| *Monthly income level* | |
| < $3,000 | 82 (21.6%) |
| $3,000–$5000 | 146 (38.6%) |
| >$5,000 | 150 (39.6%) |
| *Mother's age (year)* | |
| Mean | 34.15 |
| SD | 4.9 |
| Range | 22–48 |
| *Mother's birth place* | |
| Canada | 207 (54.8%) |
| Outside of Canada | 171 (45.2%) |
| *Child's gender* | |
| Boy | 187 (49.4%) |
| Girl | 191 (50.6%) |
| *Child's age (years)* | |
| 2 | 63 (16.6%) |
| 3 | 54 (14.2%) |
| 4 | 122 (32.2%) |
| 5 | 115 (30.4%) |
| 6 | 24 (6.3%) |
| *Child's birth place* | |
| Canada | 325 (86%) |
| Outside of Canada | 53 (14%) |
| *Child's dental insurance* | |
| No insurance | 95 (25.1%) |
| Has insurance | 283 (74.8%) |
| *Type of Insurance** | |
| Private | 247 (87.3%) |
| Public | 36 (12.7%) |
| *Toothbrushing frequency* | |
| <2x/day | 167 (42.6) |
| ≥2x/day | 211 (57.4) |
| *Sugar-intake frequency* | |
| ≥1x/day | 225 (59.5) |
| <1x/day | 153 (40.5) |
| *Utilization of dental services (last year)* | |
| No | 185 (48.9) |
| Yes | 193 (51.1) |
| *Pattern of dental attendance*** | |
| Dental problem | 31 (16.1) |
| Regular checkup | 162 (83.9) |

*Considering the 283 individuals who had dental insurance.

** Considering a total of 193 children who used dental services within the previous year.

**Table 2. Bootstrapped direct and indirect effects for the adapted TPB model.**

| Effect | β | Bootstrap SE | Bias-corrected 95% CI | p |
|---|---|---|---|---|
| *Direct effects* | | | | |
| Predisposing-enabling | 0.856 | 0.161 | 0.547/0.986 | 0.001 |
| Predisposing-attitude | -1.033 | 0.955 | -3.237/-0.340 | 0.002 |
| Predisposing-subjective norm | -0.981 | 1.102 | -3.481/-0.263 | 0.004 |
| Predisposing-PBC* | -0.986 | 1.011 | -3.509/-0.302 | 0.009 |
| Predisposing-dental attendance | -1.768 | 2.572 | -10.020/-0.333 | 0.032 |
| Predisposing-toothbrushing | -0.563 | 0.479 | -1.660/-0.223 | 0.001 |
| Predisposing-sugar intake | 0.007 | 0.173 | -0.257/0.220 | 0.312 |
| Enabling-attitude | 1.237 | 0.914 | 0.754/3.715 | 0.001 |
| Enabling-subjective norm | 1.271 | 1.039 | 0.763/4.013 | 0.001 |
| Enabling- PBC* | 1.260 | 0.964 | 0.785/4.098 | 0.001 |
| Enabling-dental attendance | 2.316 | 2.629 | 0.849/10.167 | 0.021 |
| Enabling-toothbrushing | 0.633 | 0.479 | 0.356/1.781 | 0.001 |
| Enabling-sugar intake | 0.084 | 0.161 | -0.107/0.341 | 0.422 |
| SOC-attitude | 0.353 | 0.057 | 0.258/0.444 | 0.002 |
| SOC-subjective norm | 0.236 | 0.055 | 0.136/0.314 | 0.005 |
| SOC-PBC | 0.234 | 0.063 | 0.133/0.335 | 0.002 |
| SOC-intention | -0.007 | 0.056 | -0.107/0.083 | 0.880 |
| SOC-dental attendance | 0.464 | 0.227 | 0.182/0.958 | 0.014 |
| SOC-toothbrushing | 0.097 | 0.058 | -0.004/0.189 | 0.114 |
| SOC-sugar intake | 0.084 | 0.055 | 0.072/0.255 | 0.008 |
| Attitude-intention | 0.239 | 0.059 | 0.143/0.335 | 0.002 |
| Subjective norm-intention | 0.162 | 0.059 | 0.050/0.247 | 0.012 |
| Perceived control-intention | 0.310 | 0.052 | 0.227/0.402 | 0.003 |
| Attitude-dental attendance | -0.298 | 0.278 | -0.861/0.076 | 0.166 |
| Subjective norm-dental attendance | -0.276 | 0.259 | -0.876/0.037 | 0.140 |
| PBC-dental attendance | -0.348 | 0.254 | -0.811/-0.010 | 0.084 |
| Intention-dental attendance | -0.007 | 0.067 | -0.113/0.110 | 0.929 |
| *Indirect effects* | | | | |
| Predisposing-Attitude | 1.058 | 0.984 | 0.430/3.661 | 0.001 |
| Predisposing-subjective norm | 1.088 | 1.107 | 0.414/3.886 | 0.001 |
| Predisposing- PBC | 1.078 | 1.033 | 0.455/4.045 | 0.001 |
| Predisposing-intention | 0.052 | 0.058 | -0.041/0.150 | 0.360 |
| Predisposing-dental attendance | 1.912 | 2.576 | 0.488/10.888 | 0.017 |
| Predisposing-toothbrushing | 0.542 | 0.510 | 0.207/1.857 | 0.001 |
| Predisposing-sugar intake | 0.072 | 0.152 | -0.058/0.447 | 0.289 |
| Enabling-intention | 0.892 | 0.687 | 0.550/2.646 | 0.001 |
| Enabling-dental attendance | -1.164 | 1.748 | -5.993/-0.049 | 0.086 |
| SOC-intention | 0.195 | 0.043 | 0.124/0.262 | 0.003 |
| SOC-dental attendance | -0.253 | 0.221 | -0.752/0.007 | 0.111 |
| Attitude-dental attendance | -0.002 | 0.017 | -0.024/0.030 | 0.924 |
| Subjective norm-dental attendance | -0.001 | 0.012 | -0.019/0.020 | 0.939 |
| PBC-ental attendance | -0.002 | 0.022 | -0.036/0.034 | 0.933 |

*β* = bootstrapped standardised estimate; SE = standard error; CI = confidence interval.

* Perceived Behavioural Control

family income and having dental insurance. In addition, both enabling factors and SOC were linked indirectly to behavioural intention via the three TPB components (Fig 3). It would seem that the effect of the enabling factor (greater family income and dental insurance) on parent's behavioural intention is, as would be hypothesized by the TPB model, indirectly associated with parent's perceived attitude, subjective norms and PBC towards dental attendance. Similarly, parents' behavioural intention is indirectly affected by their SOC via their perceived attitude, subjective norms and behavioural control towards dental attendance.

## Discussion

In this study, we extended the TPB model to account for parent's SOC. Using an advanced statistical technique—SEM—revealed that predisposing factors (mother and child's birthplace) significantly predicted enabling resources (family income and child's dental insurance); both factors predicted the TPB components (PBC, SN, and attitude). TPB components predicted behavioural intention; however, contrary to expectation, intention did not significantly predict dental attendance. SOC significantly predicted TPB components and dental attendance. Overall, this model explained a great deal—56%—of the variance in dental attendance in preschoolers.

Although both predisposing and enabling factors were linked to the frequency of tooth brushing; they were not significantly associated with sugar intake frequency. Mothers' SOC was the only component linked to sugary intake frequency, but it was not associated with tooth brushing behaviour. This inconsistency may imply the existence of specific contributing factors for each behaviour of interest while studying the predictors or developing interventions. Another reason for this discrepancy might be the fact that preschoolers' oral hygiene practices require additional technical support and skills from parents in addition to their SOC comparing with sugary intake frequency [23].

As for dental attendance, both predisposing and enabling factors were linked to the behaviour directly and indirectly. The significant direct link showed the independent/direct contribution of these two factors to the extended TPB model in predicting dental attendance among children. Although 74.8% of children had dental insurance and some free preventive dental services are available for children in Canada [31], less than half of the children (42%) had a preventive dental visit during the last year. This indicated the underutilization of available dental services that might be partly attributable to low parental awareness or some other barriers such as parents' time constraints or some psychosocial factors such as SOC [23, 31, 32].

SOC, an important psychosocial determinant in the oral health domain, has been applied to study the use of oral health services in a few studies [19, 23, 33]. Holde et al., for example, tested modified Andersen's behavioural model, by adding SOC construct, and found that a stronger SOC was related to more use of dental services in Norwegian adults when the association was mediated through enabling resources [33]. Among children, those whose mothers had stronger SOC scores were more likely to use dental services [19, 23] even in families with low socioeconomic status [19]. In our study, a greater SOC was directly linked to higher TPB components (perceived attitude, subjective norms and PBC) scores, and greater dental attendance; SOC was also indirectly related to behavioural intention through the TPB components. Therefore, it could be concluded that incorporating the concept of SOC into the TPB model has improved the predictability of the model by linking to the TPB components and directly to the behavior.

All TPB components in this study are linked to behavioural intention; however, the behavioural intention failed to translate into dental attendance behavior. Therefore, the TPB model itself was able to predict parents' intention to take their children for preventive dental visits

and not the actual performance of the behaviour. These findings are in line with previous studies outside of oral health domain [34, 35]. There are three concerns regarding this observation in our study. First, measurement of intention was limited to only one question/indicator measured parental intention; therefore, low variation in the items measuring intention might result in the lack of association [27]. Second, we measured mothers' self-reported past behaviours in this study, not the consecutive behaviours [36]; and third, intentions and behaviours were both measured simultaneously and no time frame existed between both measurements [34]. Therefore, longitudinal observation of performing the succeeding behaviour is required to assess the causality relationships between TPB constructs and draw more accurate conclusions [34, 35].

In this model, 34% and 56% of the variance was accounted for behavioural intention and dental attendance variables respectively. Among the predictors of intention within TPB model in our study, PBC was the strongest predictor accounted for 31% of variance to predict it followed by attitude and subjective norms with values of 24% and 16% respectively. Generally, TPB explains 20%–40% of the variance of numerous behaviours in the health domain [34, 37]. In the oral health domain, there are a few previous studies that have applied the TPB and its extended modifications to predict Oral Health Behaviours (OHB). In the study done by Buunk et al., the components of the TPB model and oral health knowledge explained 32.3% of the variance of oral hygiene behaviours including tooth brushing, flossing, and tongue cleaning among Dutch adults [17]; PBC was also the best predictor of OHB, which was in accordance with the results of a meta-analysis reporting PBC as the strongest predictor of health behaviour in the TPB model [34].

Among adolescents, Pakpour et al. tested the extended TPB model by adding action and coping planning suggesting that these two factors may help to overcome the barriers towards implementation of behavioural intention; they concluded that the expanded model accounted for 59.6% of the variance for tooth brushing behaviour. Similar to our study, they reported that perceived behavioural control was the strongest predictor of TPB in their model [38]. Dumitrescu et al. tested another extension of the TPB model, by adding oral health knowledge, among young adults and concluded that participants' attitude, PBC, and oral health knowledge predicted 51.5% of the variance to predict behavioural intention to improve tooth brushing, flossing, and dental attendance behaviours [14]; however they reported that knowledge was linked to attitude in such a way that increased knowledge led to stronger attitude, which was more stable and resistant to change [14].

In a longitudinal study using an extended version of TPB model, adult participants' attitudes, subjective norms, PBC were significant predictors of intention while participant's intention, self-efficacy and past dental attendance were significant predictors of actual dental attendance [15]. In this prospective cohort study, authors proposed "past dental attendance" as a potential predictor of individual's intention and future behaviour and hypothesized that the inclusion of past experience significantly contributed to the prediction of behavioural intention; they concluded that past behaviour predicted intention beyond TPB components. Their proposed model was able to explain 12.0% of the variance to predict intention. All four components were identified as independent predictors of individual's intention in the model. The TPB model explained 15.5% of the variance in dental attendance while adding the "past behaviour" component increased it by 7.0% [15]. In our study, we measured participants' past behaviour as their actual behaviour that might cause the absence of a link between intention and behaviour in the TPB model. Therefore, it could be recommended to design longitudinal studies to evaluate our extended TPB model to predict dental attendance behaviour prospectively.

In 2013, Van den Branden et al. in Belgium developed and validated a TPB-based questionnaire to predict parents' determinants of oral health behaviours, including dietary habits, oral hygiene, and dental attendance for children using CFA and multiple regression analysis. For dietary habits, tooth brushing, and dental attendance, TPB model accounted for 44%, 49%, and 55% of the total variance in the regression model to predict the behaviours respectively. Participants' dental attendance was predicted by both their parents' intention and PBC [11]. Among TPB components, PBC was the strongest predictor of intention which was in line with our results; however, neither intention nor PBC was significantly linked to the behaviour in our study. This inconsistency could be explained by adopting SEM analysis in our study to identify the significant pathways between TPB components while measuring model's goodness of fit and eliminating the effects of confounding variables comparing with regression analysis. SEM enabled us to control the measurement errors and achieve more accurate estimates for studied regression-coefficients.

In this study, we examined the predictability of the extended TPB model and the direct and indirect effects among the factors; however, the study was of a cross-sectional design which means no causality can be assumed. For example, the components of the TPB may lead to greater SOC or vice versa as tested in our model. Only by collecting longitudinal data in which SOC is measured at baseline, alongside, parent and child demographics, TPB components are collected at a second-time point, and finally outcomes at a third-time point can we explore cause and effect relationships. In this research, only one item adopted to measure intention; therefore, having more items to assess this construct in the future studies will enhance the internal validity of the questionnaire and reduce measurement errors. A further limitation was the use of a convenience sampling method and self-reported outcome variables. For example, dental attendance frequency may have been over-estimated as parents may have answered the question according to how often they should be taking their child to their dentist. Based on the available data, the vaccination rate in Edmonton in 2013–14 was approximately 91%. Nonetheless, it's recommended to include mothers who refuse immunizations for their children, as they may have different views about taking their children to receive preventive care services, such as regular dental check-ups. Finally, the external validity of the extended TPB model needs to be investigated in other population groups such as adolescents, young adults, and adults in the future studies.

## Conclusions

- Predisposing factors significantly predicted enabling resources; both predicted the TPB components (attitude, subjective norms, and PBC).

- TPB components predicted behavioural intention; however, nor intention neither PBC significantly predicted dental attendance.

- SOC was directly linked to TPB components and dental attendance while indirectly related to behavioural intention through the TPB components.

- Overall, 56% of the variance in dental attendance was explained by the expanded TPB model.

## Acknowledgments

The authors wish to thank the Alberta Health Services research coordinators, community health centers' managers and staff, and parents participated in the study.

## Author Contributions

**Conceptualization:** Maryam Elyasi, Hollis Lai, Paul W. Major, Sarah R. Baker, Maryam Amin.

**Data curation:** Maryam Elyasi.

**Formal analysis:** Maryam Elyasi, Hollis Lai, Sarah R. Baker.

**Funding acquisition:** Maryam Elyasi, Maryam Amin.

**Investigation:** Maryam Elyasi, Maryam Amin.

**Methodology:** Maryam Elyasi, Hollis Lai, Paul W. Major, Sarah R. Baker, Maryam Amin.

**Writing – original draft:** Maryam Elyasi, Sarah R. Baker.

**Writing – review & editing:** Maryam Elyasi, Hollis Lai, Paul W. Major, Sarah R. Baker, Maryam Amin.

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
