## [Decision Letter · Decision Letter 0]

23 Oct 2019

PONE-D-19-26262

Modeling the Theory of Planned Behaviour to predict adherence to preventive dental visits in preschool children

PLOS ONE

Dear Dr. Amin,

Thank you for submitting your manuscript to PLOS ONE. After careful consideration, we feel that it has merit but does not fully meet PLOS ONE’s publication criteria as it currently stands. Therefore, we invite you to submit a revised version of the manuscript that addresses the points raised during the review process.

We would appreciate receiving your revised manuscript by Dec 07 2019 11:59PM. To enhance the reproducibility of your results, we recommend that if applicable you deposit your laboratory protocols in protocols.io, where a protocol can be assigned its own identifier (DOI) such that it can be cited independently in the future. For instructions see: http://journals.plos.org/plosone/s/submission-guidelines#loc-laboratory-protocols

We look forward to receiving your revised manuscript.

Kind regards,

Frédéric Denis, Ph.D.

Academic Editor

PLOS ONE

**Journal Requirements:**

**Comments to the Author**

1. Is the manuscript technically sound, and do the data support the conclusions?

Reviewer #1: Yes

Reviewer #2: Yes

2. Has the statistical analysis been performed appropriately and rigorously? 

Reviewer #1: Yes

Reviewer #2: Yes

3. Have the authors made all data underlying the findings in their manuscript fully available?

Reviewer #1: Yes

Reviewer #2: Yes

4. Is the manuscript presented in an intelligible fashion and written in standard English?

Reviewer #1: Yes

Reviewer #2: Yes

5. Review Comments to the Author

Reviewer #1: Major points

-Figure 2 is difficult to understand and needs to be re-done. Consider breaking the information up into two graphics, adding a color code, and/or move the significant figures to a legend. As it is, arrows overlap the shapes, the shapes are arranged without a discernable pattern, and the significant figures are small and hard to read and it is difficult to determine which metrics go with each interaction effect.

-I worry about bias occurring in the study results. By being at the health center, parents were actively engaged in attaining health care for their children and therefore might not be representative of the greater population. Do you have any studies to confirm that 42% a parents seeking dental care visits is close to the average rate. Likewise, is it possible that the context in which the subjects were questioned ( a community health centers ) may have influence healthcare related intentional behavior? Were subjects in a similar context(s) in the other studies you cite that extend the TPB models to: adolescents, young adults, and adults? I would like to see both of these points addressed in the discussion as potential limitations.

- Please explain in more detail how an subject’s dental attendance metric was derived from their Oral Health behaviors survey results (sub-section ‘d’ from the Materials and Methods section) – likewise please include the descriptive statistics on these answers in Table 1.

Minor points

-On lines 324 to 325 you say ‘…83% 325 of children had dental insurance…’ but in Table 1 you report that 74.8% of children had dental insurance – please fix the discrepancy.

-The sentence fragments below need to be re-worded (lines 341-343) –

One surprising finding was from studying the TPB components. While they were all linked to a higher behavioural intention. However, the behavioural intention was not associated with greater dental attendance.

Reviewer #2: Dear Editorial Manager

Thank you for having choosing me as a reviewer.

• This article is very interesting because it deals with an essential topic : how to predict a healthy behaviour (e.g. preventive visits at the dental office) and how to explain the transition between the intention and the actual attendance.

• This article is modeling the true life of our patients : they think they (or their children) must visit the dentist, they want but the actual attendance has never happened or later on. We can read that even the possibility of free preventive visits is underutilized. Is it due to the age of the children ? But as AAPD recommends, the awareness of oral health begins at birth. Do the AAPD recommendations are known by parents ?

• This article shows that SOC is very important in the decision. Using SOC for extending the TPB is a good manner to approach the understanding of behaviours related to preschoolers’ oral health and particularly dental attendance.

• Knowledge may influence also these oral health behaviours. In a future study, may be a questionnaire on children’ oral health : importance and roles of temporary teeth, formation of the oral microbiote and its consequences etc....

• Just a minor revision : lines 165 and 166 : the results of the Cronbach’s alpha must be in the section « Results ». It’s already written in lines 234 and 235.

6. PLOS authors have the option to publish the peer review history of their article (what does this mean?). If published, this will include your full peer review and any attached files.

Reviewer #1: No

Reviewer #2: Yes: Javotte NANCY

---

## [Author Response · Author response to Decision Letter 0]

6 Dec 2019

Reviewer: #1

Major points

“Figure 2 is difficult to understand and needs to be re-done. Consider breaking the information up into two graphics, adding a color code, and/or move the significant figures to a legend. As it is, arrows overlap the shapes, the shapes are arranged without a discernable pattern, and the significant figures are small and hard to read and it is difficult to determine which metrics go with each interaction effect.”

Response: We appreciate the reviewer’s constructive suggestion. The information provided in Figure 2 is represented in two separate figures, New Figure 2 and Figure 3, to demonstrate the direct and indirect relationships between variables distinctly. Adjustments were made to eliminate overlaps, arrange the shapes, and enlarge the figures. Also, the metrics applied to evaluate the interaction effects were clarified in new figures. 

“I worry about bias occurring in the study results. By being at the health center, parents were actively engaged in attaining health care for their children and therefore might not be representative of the greater population. Do you have any studies to confirm that 42% a parents seeking dental care visits is close to the average rate? Likewise, is it possible that the context in which the subjects were questioned (a community health centers) may have influence healthcare related intentional behavior? Were subjects in a similar context(s) in the other studies you cite that extend the TPB models to: adolescents, young adults, and adults? I would like to see both of these points addressed in the discussion as potential limitations.”

Response: The reviewer’s abovementioned important points were addressed in the Methods and Discussion sections. The parents were recruited through vaccination programs that hold only at community health centers in Edmonton, AB. According to the 2013-14 Alberta Health Services Report, the overall vaccination rate for preschoolers in Edmonton in 2014 was 91.7%. Therefore, approaching community health centers enabled us to recruit a good representative of our target population. Regarding the reported 42% of dental attendance among pre-schoolers in our study, there is another study done by Locker et al. (Ref #26) reported 42.8% of the regular dental visits among immigrant children and 72.6% among non-immigrant children. Although the age group in these two studies are different but our reported percentage was not comparably high. 

The following statements has added to the Methods and Discussion sections respectively:

- Page 7, lines142-143: “According to the 2013-14 Alberta Health Services Report, the overall vaccination rate for preschoolers in Edmonton was 91.7%.”

- Page 20, lines 411-416: “Based on the available data, the vaccination rate in Edmonton in 2013-14 was approximately 91%. Nonetheless, it’s recommended to include mothers who refuse immunizations for their children, as they may have different views about taking their children to receive preventive care services, such as regular dental check-ups. Finally, the external validity of the extended TPB model needs to be investigated in other population groups such as adolescents, young adults, and adults in the future studies.”

“Please explain in more detail how a subject’s dental attendance metric was derived from their Oral Health behaviors survey results (sub-section‘d’ from the Materials and Methods section) – likewise please include the descriptive statistics on these answers in Table 1.”

Response: The required changes were made in the body of the manuscript (pages 8-9, lines 174-182) and table 1 (pages 11-12) accordingly. 

Minor points

“On lines 324 to 325 you say ‘…83% 325 of children had dental insurance…’ but in Table 1 you report that 74.8% of children had dental insurance – please fix the discrepancy.” 

Response: Thanks for the observation; the percentage reported in Table 1 was corrected. The abovementioned discrepancy has been fixed on page 16, line 325.

“The sentence fragments below need to be re-worded (lines 341-343: 

One surprising finding was from studying the TPB components. While they were all linked to a higher behavioural intention. However, the behavioural intention was not associated with greater dental attendance.” 

Response: The abovementioned statement was revised on page 17, lines 341-342:

“All TPB components in this study are linked to behavioural intention; however, the behavioural intention failed to translate into dental attendance behaviour.”

Reviewer: #2

• This article is very interesting because it deals with an essential topic: how to predict a healthy behaviour (e.g. preventive visits at the dental office) and how to explain the transition between the intention and the actual attendance. 

This article is modeling the true life of our patients: they think they (or their children) must visit the dentist, they want but the actual attendance has never happened or later on. We can read that even the possibility of free preventive visits is underutilized. Is it due to the age of the children? But as AAPD recommends, the awareness of oral health begins at birth. Do the AAPD recommendations are known by parents?

• This article shows that SOC is very important in the decision. Using SOC for extending the TPB is a good manner to approach the understanding of behaviours related to preschoolers’ oral health and particularly dental attendance.

• Knowledge may influence also these oral health behaviours. In a future study, may be a questionnaire on children’ oral health: importance and roles of temporary teeth, formation of the oral microbiote and its consequences etc....

Response: We appreciate the reviewer’s thoughtful feedback that will help us design and develop our future studies.

• Just a minor revision: lines 165 and 166: the results of the Cronbach’s alpha must be in the section « Results ». It’s already written in lines 234 and 235.

Response: We acknowledge the reviewer’s remark. The results of Cronbach’s alpha removed from the Method’s section (line 166).

---

## [Editor Report · Decision Letter 1]

16 Dec 2019

Modeling the Theory of Planned Behaviour to predict adherence to preventive dental visits in preschool children

PONE-D-19-26262R1

Dear Dr. Amin,

We are pleased to inform you that your manuscript has been judged scientifically suitable for publication and will be formally accepted for publication once it complies with all outstanding technical requirements.

With kind regards,

Frédéric Denis, Ph.D.

Academic Editor

PLOS ONE
---

## [Editor Report · Acceptance letter]

26 Dec 2019

PONE-D-19-26262R1 

Modeling the Theory of Planned Behaviour to predict adherence to preventive dental visits in preschool children 

Dear Dr. Amin:

I am pleased to inform you that your manuscript has been deemed suitable for publication in PLOS ONE. Congratulations! Your manuscript is now with our production department. 

With kind regards,

on behalf of

Dr. Frédéric Denis 

Academic Editor

PLOS ONE